# Multiple phylogenetically-diverse, differentially-virulent *Burkholderia pseudomallei* isolated from a single soil sample collected in Thailand

**Chandler Roe**[1], **Adam J. Vazquez**[1], **Paul D. Phillips**[1], **Chris J. Allender**[1¤a], **Richard A. Bowen**[2], **Roxanne D. Nottingham**[1], **Adina Doyle**[1], **Gumphol Wongsuwan**[3], **Vanaporn Wuthiekanun**[3], **Direk Limmathurotsakul**[3], **Sharon Peacock**[4], **Paul Keim**[1], **Apichai Tuanyok**[1¤b], **David M. Wagner**[1], **Jason W. Sahl**[1] *

1 The Pathogen and Microbiome Institute, Northern Arizona University, Flagstaff, Arizona, United States of America, 2 Department of Biological Sciences, Colorado State University, Ft. Collins, Colorado, United States of America, 3 Mahidol-Oxford Tropical Medicine Research Unit, Mahidol University, Bangkok, Thailand, 4 Department of Medicine, University of Cambridge, Cambridge, England

¤a Current address: Department of Pathogen Genomics, Translational Genomics Research Institute, Flagstaff, Arizona, United States of America
¤b Current address: Department of Infectious Diseases and Immunology, College of Veterinary Medicine; and Emerging Pathogens Institute, University of Florida, Gainesville, Florida, United States of America
* Jason.Sahl@nau.edu

**Data Availability Statement:** All genomic data are deposited under BioProject number

## Abstract

*Burkholderia pseudomallei* is a soil-dwelling bacterium endemic to Southeast Asia and northern Australia that causes the disease, melioidosis. Although the global genomic diversity of clinical *B. pseudomallei* isolates has been investigated, there is limited understanding of its genomic diversity across small geographic scales, especially in soil. In this study, we obtained 288 *B. pseudomallei* isolates from a single soil sample (~100g; intensive site 2, INT2) collected at a depth of 30cm from a site in Ubon Ratchathani Province, Thailand. We sequenced the genomes of 169 of these isolates that represent 7 distinct sequence types (STs), including a new ST (ST1820), based on multi-locus sequence typing (MLST) analysis. A core genome SNP phylogeny demonstrated that all identified STs share a recent common ancestor that diverged an estimated 796–1260 years ago. A pan-genomics analysis demonstrated recombination between clades and intra-MLST phylogenetic and gene differences. To identify potential differential virulence between STs, groups of BALB/c mice (5 mice/isolate) were challenged via subcutaneous injection (500 CFUs) with 30 INT2 isolates representing 5 different STs; over the 21-day experiment, eight isolates killed all mice, 2 isolates killed an intermediate number of mice (1–2), and 20 isolates killed no mice. Although the virulence results were largely stratified by ST, one virulent isolate and six attenuated isolates were from the same ST (ST1005), suggesting that variably conserved genomic regions may contribute to virulence. Genomes from the animal-challenged isolates were subjected to a bacterial genome-wide association study to identify genomic regions associated with differential virulence. One associated region is a unique variant of Hcp1, a component of the type VI secretion system, which may result in attenuation. The results of this

PRJNA429426; specific accession numbers are shown in S1 Table.

**Funding:** The authors received no specific funding for this work.

**Competing interests:** The authors have declared that no competing interests exist.

study have implications for comprehensive sampling strategies, environmental exposure risk assessment, and understanding recombination and differential virulence in *B. pseudomallei*.

## Author summary

*Burkholderia pseudomallei* is the causative agent of melioidosis, a disease endemic to Southeast Asia and northern Australia. The evaluation of diversity within *B. pseudomallei* has largely been conducted utilizing clinical isolates despite almost all infections emerging from environmental exposure. In this study, we surveyed the genomic diversity of 169 isolates collected from a single soil sample in Ubon Ratchathani Province, Thailand. Seven different sequence types were identified, and substantial within-sequence-type gene diversity was observed. To test for differential virulence, 30 isolates were challenged in a mouse melioidosis model. A small number of isolates killed all mice, but most killed none, demonstrating the variable virulence potential of different *B. pseudomallei* isolates present in the single sample. A comparative genomics analysis identified multiple genes associated with virulence, including Hcp1, a component of the type VI secretion system and a known virulence factor. The results of this study have implications for the comprehensive environmental surveillance, environmental exposure, and differential virulence of *B. pseudomallei*.

## Introduction

*Burkholderia pseudomallei* is a soil-dwelling bacterium endemic to Southeast Asia and northern Australia where it is the causative agent of melioidosis, a potentially fatal disease in humans [1]. In Thailand, *B. pseudomallei* causes 19% of community-acquired bacteremia [2] and in northern Australia, melioidosis is the most common cause of fatal bacteremic pneumonia [3]. Agricultural workers are among the highest at-risk population to develop infection due to repeated exposure to *B. pseudomallei*, primarily due to direct contact with compromised skin [4]. Melioidosis can be difficult to treat with antibiotics [5] and has been associated with a mortality rate as high as 50% in symptomatic individuals [6]. The most probable route of infection occurs through percutaneous inoculation [7], although inhalation and ingestion are also important [8,9].

Most genome sequencing of *B. pseudomallei* has focused on clinical isolates, although sequencing and analysis of environmental isolates has been performed, including recent studies focused on novel diversity in the Caribbean [10,11]. Other studies have investigated the microdiversity of *B. pseudomallei* in the environment using sub-genomic methods, such as pulsed-field gel electrophoresis (PFGE) [12] and multi-locus sequence typing (MLST) [13,14]. Although one study found 4 distinct sequence types (STs) within a single soil sample in Ubon, Thailand [12], the resulting data did not identify genomic and phylogenetic differences within each ST. Another study surveyed genotypic diversity with sub-genomic methods in Northeastern Thailand and found 7 distinct STs across 11 soil samples [13]. A study on MLST types in Australia identified that the diversity of *B. pseudomallei* populations increased with sampling area, suggesting localized adaptation [15]. A recent study used whole genome sequencing to identify genomic signatures that differ between clinical and environmental *B. pseudomallei* isolates in Northeastern Thailand [16]; however, the focus of this study was the overlap

between clinical and environmental isolates and not the within-ST diversity of *B. pseudomallei*.

The *B. pseudomallei* genome is highly plastic [17], owing largely to the acquisition of genomic islands (GIs) [18], which are often acquired from other species via horizontal gene transfer [18]. These GIs generally consist of several contiguous genes and account for an average 5.8% of an individual *B. pseudomallei* genome [19]. Previous studies have demonstrated high levels of homologous recombination throughout the *B. pseudomallei* genome, including within housekeeping genes [20]; research has shown that recombination is the major driver of evolution in *B. pseudomallei* [19] and a diverse and adaptable genome is crucial for its environmental adaptation and success within soil [14,21,22]. Understanding the accessory genome of *B. pseudomallei* isolates collected from a limited geographic region may provide a better understanding of its ability to adapt to local ecological conditions.

Multiple mechanisms associated with pathogenesis and virulence have been described in *B. pseudomallei*. One of the most characterized virulence factors is the cluster 1 type VI secretion system (T6SS) [23], although the type III secretion system cluster 3 also has been associated with virulence [9]. Recent studies associated a filamentous hemagglutinin (FhaB3) with septic shock and mortality in Australia [24] and the oxidoreductase membrane protein DsbB with virulence based on a knock-out study and subsequent animal challenge [25]. Although the characterization of these factors has improved our understanding of *B. pseudomallei* pathogenesis, most mechanisms remain unidentified and virulence in this species remains poorly understood [26,27]. These types of analyses are further confounded by host factors associated with melioidosis [28], suggesting that an interplay of pathogen and host factors are associated with disease severity and patient outcome [8].

The purpose of this study was to examine the evolutionary dynamics of environmental *B. pseudomallei* and investigate differential virulence from a single soil sample from Thailand. We sequenced 169 *B. pseudomallei* genomes from the soil sample out of a set of 288 isolates. A total of 30 isolates were selected for a murine infection study and comparative genomics and machine learning were used to identify differences between differentially-virulent groups. The results have implications for environmental sampling strategies for *B. pseudomallei* that are focused on understanding the virulence potential of clinical and environmental isolates. The characterization of genomic diversity and virulence of sympatric isolates will aid in the understanding of the evolution, ecology, and disease potential of this important human pathogen.

## Methods

### Ethics statement

Mouse challenge experiments were approved by the Animal Care and Use Committee of Colorado State University (CSU), protocol number 17-7497A. The experiments were performed under Select Agent and ABSL-3 containment practices at CSU and in strict accordance with the recommendations in the Guide for the Care and Use of Laboratory Animals of the National Institutes of Health. Every effort was made to minimize animal suffering and pain.

**B. pseudomallei isolation.** A single site was chosen in the Ubon Rachathani Province (a.k.a. Ubon) of Thailand for sample collection (Fig 1A). The site, a rice field, was sampled in the dry season in a region where *B. pseudomallei* is highly endemic. The map of Thailand was created using ArcGIS software by Esri. ArcGIS and ArcMap are the intellectual property of Esri and are used herein under license, all rights reserved. The site was divided into a grid of 50 holes, each separated by 2m (Fig 1B). On 2 April 2007, soil was collected from all 50 holes at a depth of both 10cm and 30cm with a total of 100g of soil collected per sample; we report the results from just one of these 100 samples here as it yielded the largest number of *B*.

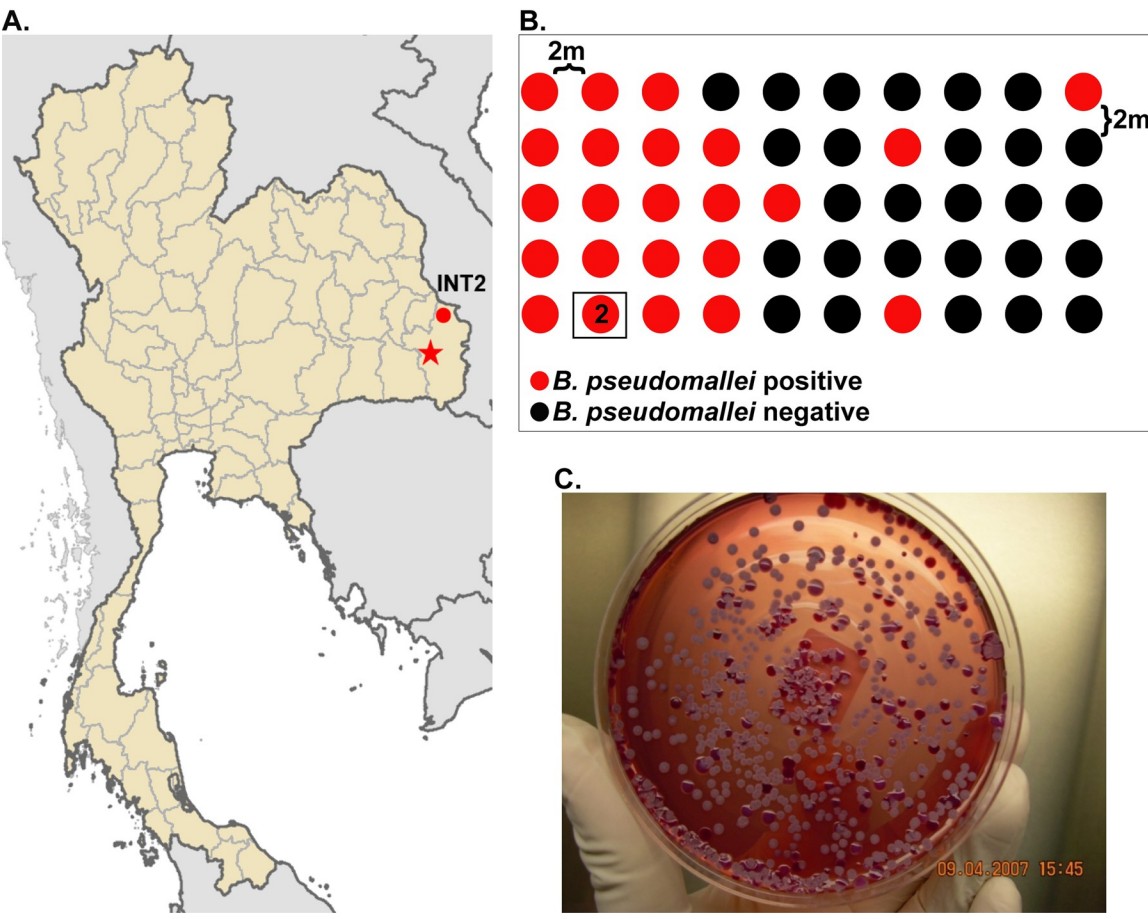

**Fig 1. The sampling strategy performed in this study.** A) The location in Thailand (circle) where samples were obtained; the city of Ubon Ratchathani within Ubon Ratchathani province is marked by the star. The map of Thailand was created using ArcGIS software by Esri. ArcGIS and ArcMap are the intellectual property of Esri and are used herein under license, all rights reserved, B) the sampling strategy at the intensive sampling site; the location where the INT2 sample was collected is marked and boxed, C) a plate showing the *B. pseudomallei* colonies grown out of the single enriched INT2 soil sample; a total of 288 isolates were picked from this plate, and 169 of these isolates were sequenced.

*pseudomallei* isolates. All soil samples were mixed with 100ml of sterile distilled water in zipped bags and the bags were left to stand at room temperature overnight. A direct plating technique was used to isolate *B. pseudomallei* by inoculating 2 different volumes of the soil solution, 10 and 100μl, onto an Ashdown agar plate. The inoculum was spread evenly on the agar plate with an L-shape spreader while the plate was being spun on a plate rotator. The plates were incubated at 42˚C for 7 days and were checked daily for growth. A latex agglutination test [29] was used as a presumptive test for *B. pseudomallei* identification from ~300 isolates collected from INT2 (Fig 1C). Each suspected *B. pseudomallei* colony was picked by a sterile loop and inoculated into each well of a 96-well plate containing 100μl of LB agar. The plate was sealed with adhesive aluminum foil and stored at room temperature before being shipped to Northern Arizona University under a CDC/APHIS permit. Isolates were picked, added to 70% glycerol, and stored at -80˚C.

**DNA extraction and sequencing.** Isolates were grown on LB agar at 37˚C for 24–48 hours and DNA was extracted from each isolate using the Qiagen DNeasy Blood and Tissue Extraction kit (Qiagen, Hilden, Germany) following the Gram-positive protocol with the

addition of 1mg/ml of lysozyme and doubling the volumes. DNA was prepared for multiplexed, paired-end sequencing with a 500 base pair insert using Standard PCR Library Amplification (KAPA Biosystems, Woburn, MA). Genomes were sequenced on either a 2x100 bp paired-end HiSeq run or a 2x250 bp MiSeq run.

**Genomic assembly and *in silico* genotyping.** Illumina sequence data were assembled with the SPAdes assembler v.3.10.0 [30]. The per contig coverage was determined by mapping reads against contigs with Minimap2 v2.17 [31] and calculating coverage with Samtools v1.9 [32]. Each genome assembly was manually edited to remove contigs that had an anomalously low coverage compared to the rest of the assembly or aligned against known contaminants based on a BLASTN alignment [33] against the GenBank [34] nt database. Assembly statistics and accession numbers are shown in S1 Table. The multi-locus sequence type (MLST) for each isolate was calculated from each genome sequence with stringMLST [35] (S1 Table). Representative genome assemblies were submitted to GenBank (S1 Table).

**Comparative pan-genomics.** All genome assemblies were annotated with Prokka v1.14.6 [36]. The pan- and core-genomes were calculated for each ST with Panaroo v1.2.3 [37]. The resulting pan-genome representative sequences were mapped against genome assemblies with the large-scale BLAST score ratio (LS-BSR) tool v1.2.2 [38] in conjunction with BLAT v36x2 [39]. Variably-conserved genes, based on blast score ratio (BSR) values [40], were mapped against phylogenies from each ST with LS-BSR/BLAT and visualized with the interactive tree of life (iTOL) [41].

**Single nucleotide polymorphism (SNP) identification and phylogenetics.** To understand the placement of INT2 genomes within the global *B. pseudomallei* phylogeny, INT2 genome assemblies were combined with a global set of *B. pseudomallei* genomes (S2 Table); genomes were downloaded using the ncbi-genome-download tool (https://github.com/kblin/ncbi-genome-download) on December 13, 2019. SNPs were called with nucmer v3.1 [42] in conjunction with NASP v1.0.2 [43] for genome assemblies using *B. pseudomallei* K96243 (accession number GCA_000011545.1) [44] as the reference. A phylogeny was inferred on the concatenated SNP alignment with FastTree2 v2.1.10 [45]. For all other phylogenies, raw reads were aligned against *B. pseudomallei* K96243 with Minimap2 and SNPs were called using the HaplotypeCaller method in GATK v4.1.2 [46,47]. These functions were wrapped with NASP and positions were filtered if the coverage was <10X or the minimum allele frequency ratio was <0.90. All remaining polymorphic positions were concatenated into a single multi-FASTA file. Phylogenetic trees were inferred on concatenated SNP alignments with a maximum-likelihood algorithm implemented in IQ-TREE v1.6.1 [48] using the integrate model search method [49]. The Retention Index of SNP alignments, which provides information on the extent of homoplasy, was calculated with Phangorn v2.4 [50]. The functional annotation of SNPs was determined with SnpEff v4.3t [51]. Homoplastic SNPs were identified from the FASTA file with HomoplasyFinder [52].

**Population structure.** The NASP SNP matrix including the 169 *B. pseudomallei* genomes sequenced in this study was used as input into fastSTRUCTURE v1.0 [53] in order to investigate population structure. Using a variational Bayesian framework, the most likely number of populations as well as the probability that each strain belonged to each population was calculated. Only bi-allelic sites were included in this analysis and no phenotype data were provided for this sample set. The admixture model was applied assuming SNPs were unlinked. Individual ancestry and correlated allele frequencies were simulated between the range of K = 2 to K = 10 populations. Individual marginal likelihood values from each K simulation were compared to infer the most likely number of populations (K with the lowest likelihood) using the chooseK.py script included with fastSTRUCTURE [53].

**Beast timing analysis.** BEAST analysis included all 169 genomes sequenced in this study. Putative recombinant SNPs were identified and removed with Gubbins v2.4.1 [54] prior to running BEAST. A SNP matrix based on *B. pseudomallei* chromosome I (BX571965.1) that included 5,252 total SNPs with 3,059 parsimony-informative SNPs was analyzed within a Bayesian framework separate from chromosome II (BX571966.1) data based on previous research [55]. The chromosome II dataset included 3,760 total SNPs with 2,484 informative SNPs used in the Bayesian analysis. Additionally, each ST was run through BEAST separately as well using a previous reported evolutionary rate for chromosome I and chromosome II ($9.22E^{-7}$ and $6.71E^{-7}$ substitutions per site per year, respectively) [55]. To estimate the most recent common ancestor (TMRCA) for the INT2 isolates, a Bayesian strict clock was applied in BEAST version 1.8.4 [56]. MEGA7 [57] was used to infer the best nucleotide substitution model using the Bayesian information criterion (*BIC*) (GTR for chromosome I data and HKY for the chromosome II dataset). A correction for invariant sites was implemented in all BEAST analyses by specifying a Constant Patterns model. Additionally, a "path and stepping stone" sampling marginal-likelihood estimator was used to determine best-fitting clock (strict and relaxed) and demographic model combination (Constant, Bayesian Skygrid, Bayesian Skyline, Extended Bayesian Skyline Plot, GMRF) from ten clock and model combinations [58]. However, no appreciable difference of the marginal likelihood calculation was observed. We implemented the strict skyline clock and skyline demographic model combination to avoid overparameterization. Four independent chains of 1 billion iterations were run for the strict clock and skyline demographic model combination. The program Tracer v1.6.0 [59] was used to visually confirm convergence.

***In vivo* animal challenge.** Female BALB/c mice, 6–8 weeks old, were purchased from Charles River laboratories and housed at Colorado State University. To assess virulence of a subset of 30 of the soil isolates collected from INT2, each isolate was grown at 37˚C in BHI medium to an $OD_{600}$ of 1, supplemented with glycerol to 10% v/v, and frozen in multiple aliquots at -80˚C. A vial of these stocks was thawed and serial dilutions plated on BHI agar plates to determine titer 2 days prior to the animal challenge. On the day of the animal challenge, a new frozen aliquot was thawed and diluted with PBS to obtain approximately 500 CFU per 50μL of the bacterial inoculum. The development of the subcutaneous (SC) model in BALB/c mice found an $LD_{50}$ of 1000 CFU [60]; this guided the use of 0.5x $LD_{50}$ in this study to identify differential virulence. Each mouse (5 mice per strain) was subcutaneously inoculated in the medial right leg with 50μL of the bacterial inoculum; subcutaneous infection in mouse models represents percutaneous human infections, which are common in endemic areas [61]. The same volume was also back-titrated to confirm the given dose. In addition to INT2 isolates, mice were also challenged with control strains, NCTC13178 and NCTC13179 [60], at the same dose; strains were purchased from the National Collection of Type Culture (NCTC) in the UK. Mice were weighed and evaluated for clinical symptoms associated with *B. pseudomallei* for 21 days post infection. This monitoring was performed daily for the first 7 days and subsequently every 3–4 days to 21 days. The clinical scores indicated in the study were: 0 = normal, 1 = questionable illness, 2 = mild but definitive illness, 3 = moderate illness, 4 = severe illness, moribund-euthanized, 5 = found dead. The moribund and surviving mice at 21 days post infection were euthanized by a standard $CO_2$ overdosing method. To compare the virulence of selected isolates, combined survival curves for each ST were graphed using the program GraphPad Prism version 7.00 for mac (GraphPad Software, La Jolla California, USA).

**Comparative genomics of animal-challenged isolates.** Genomes from virulent (*n* = 8) (killed all mice) and attenuated (*n* = 20) (killed no mice) strains, as determined by the mouse challenge, were processed with LS-BSR using BLAT to align coding region sequences (CDS) predicted by Prokka and clustered with CD-HIT v4.8.1 [62]; intermediate virulence strains

(*n* = 2) were not included in this analysis. CDSs were identified that were more conserved in either the virulent or attenuated phenotypes with the compare_BSR.py script in the LS-BSR repository based on BSR values; the differential conservation of genes was verified thorough short read mapping approaches where the breadth of coverage was calculated with Samtools. For comparison, the pan-genome was also determined with Panaroo using Prokka-annotated genomes.

**Genome-wide association analysis (GWAS) with pyseer.**   To identify genomic regions associated with differential virulence, a Kmer-based GWAS analysis was conducted across animal-challenged genomes. Briefly, assembled genomes were fragmented into 54 base pair Kmers using bbmap (https://sourceforge.net/projects/bbmap/) and Kmers with a frequency <4X coverage in any one genome were converted to "0"; only Kmers that were present in at least one genome were processed further. Kmers were then analyzed with pyseer [63] using default settings with phenotypic data represented by "1" for virulent and "0" for attenuated isolates. For the investigation of genes that contain associated Kmers, regions were extracted from BLASTN alignments, aligned with MUSCLE v3.8.31 [64], and visualized with Jalview [65].

**Machine learning approaches for feature identification.**   In order to identify genes associated with virulence, a machine learning algorithm (Boruta) [66] was applied to a binary LS-BSR matrix; a BSR value ≥0.8 for a CDS was coded as present ("1"), whereas a score <0.8 for a CDS was coded as absent ("0"). Of the 7 parameters the Boruta_py module (https://github.com/scikit-learn-contrib/boruta_py) requires, all were kept at default except the estimator object, which is a supervised learning model in which feature importance can be calculated. The percentage strength of Boruta's shadow features compared to the real features was also calculated, which was set to 99 instead of the default 100 to identify more potential associations. A random forest classifier from the sci-kit learn [67] module was tuned over 250 iterations of stratified 5-fold cross-validation with gridsearchCV and chosen for the estimator object. Finally, the Boruta algorithm was run for 125 iterations over the random seed parameter of the Boruta_py module on the tuned random forest model. Regions of interest were identified by only investigating features that were selected in 100% of the Boruta iterations.

**Screen of previously-characterized virulence genes.**   The peptide sequences of 551 genes previously associated with virulence in *B. pseudomallei* [68] were screened against the 30 animal-challenged genomes with TBLASTN [69] in conjunction with LS-BSR.

## Results

The diversity of *B. pseudomallei* within a single soil sample (intensive site 2, INT2) from the Ubon Ratchathani Province of Thailand was investigated by whole genome sequencing and comparative genomics. An analysis of 169 genomes demonstrated that seven different sequence types (STs) were observed, although three of the STs were single-locus variants of STs observed in INT2 genomes (S1 Table). The differential virulence of a subset of 30 of these isolates was investigated in a murine melioidosis model.

## Global phylogenetics

To understand the global diversity of sequenced genomes, the 169 INT2 genomes were compared to 1,576 globally-diverse *B. pseudomallei* genomes available in GenBank (S2 Table). The core genome phylogeny demonstrated that genomes from this study shared a recent common ancestor (Fig 2), suggesting a common geographic origin. The retention index (RI) for the concatenated SNP alignment and phylogeny was calculated at 0.8378, indicating homoplasy throughout the *B. pseudomallei* core genome, which was expected based on previous studies [70].

   

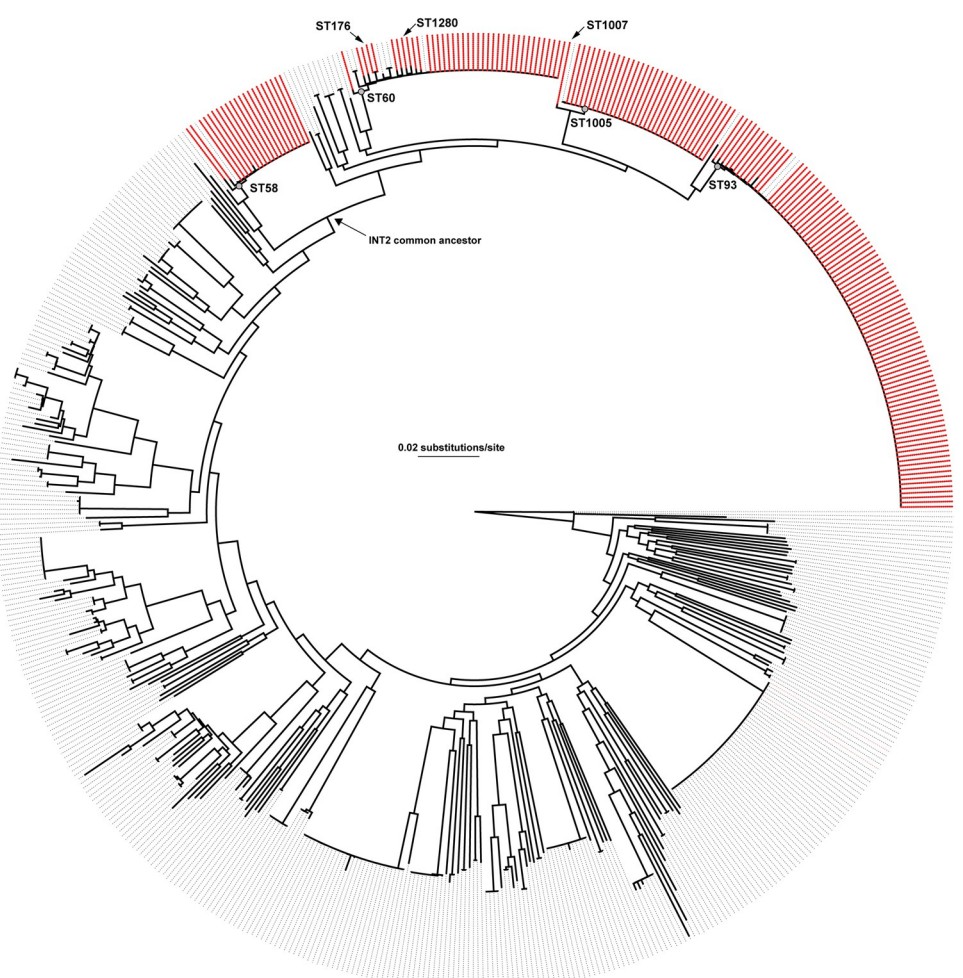

**Fig 2. An approximate maximum-likelihood phylogenetic tree inferred with FastTree2 [45] from a concatenation of 90,518 core genome SNPs called from 169 genomes sequenced in this study (red text) as well as 1,576 reference *B. pseudomallei* genomes from GenBank (S2 Table).** Genomes sequenced in this study are shown in red. The tree was rooted with MSHR668 based on its basal position in other studies [71].

## Population genomics of INT2 genomes

To better understand their genetic backgrounds, we determined the population structure of the 169 *B. pseudomallei* genomes from INT2 using Bayesian methods implemented in fastSTRUCTURE. By comparing individual marginal likelihood values to each individual K value (signifying number of populations), the most reliable population distribution was determined. The marginal likelihood values identified K = 4 as the best scenario with four distinct populations largely corresponding to ST58, ST93, ST1005, and ST60 (Fig 3); these populations are highly correlated with phylogenetic groupings (Figs 2 and 3). The ST58 population is composed of a mostly unique clade (solid yellow) with isolates Bp1763 and Bp1752 representing recombination with the ST93 clade. The ST60 population is composed of admixtures between genomes from ST93, ST1005, and ST58 populations. The ST1005 population is represented by a unique genetic clade (solid red) with only one genome (Bp1818; ST1007) displaying an admixture. Furthermore, while ST93 has the largest number of genomes (n = 82), only three isolates displayed an admixture (Bp1740, Bp1805, Bp8880).

   

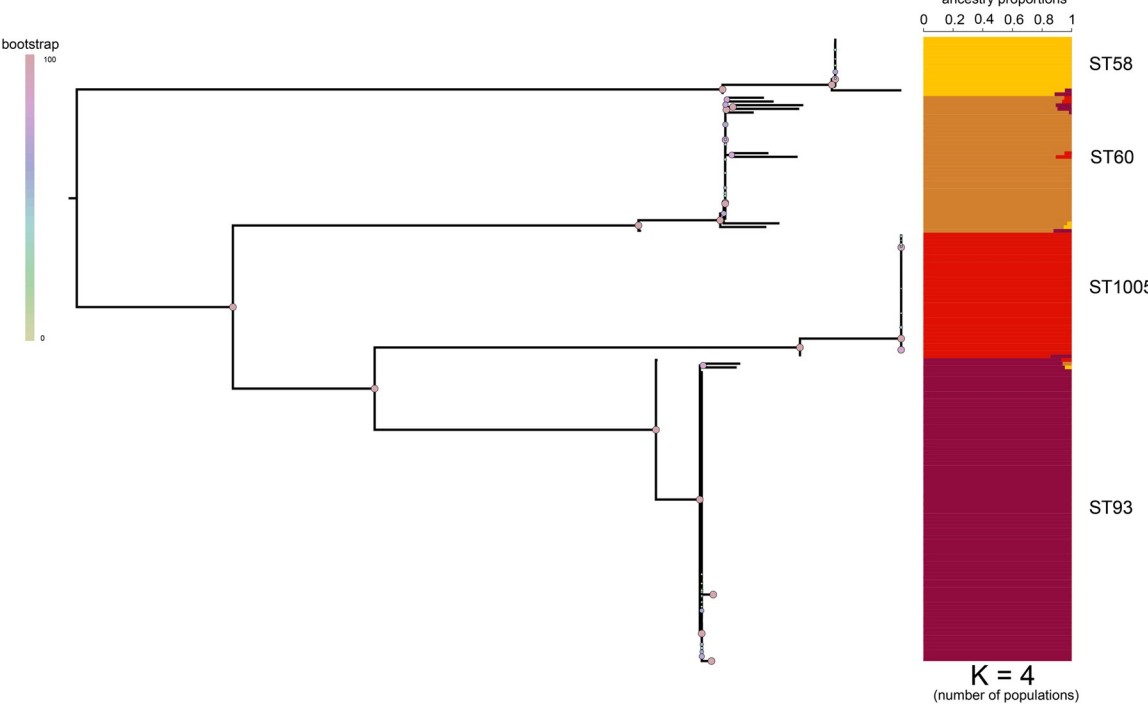

**Fig 3. The population structure of INT2 genomes, as revealed by fastSTRUCTURE [53].** The maximum likelihood tree was inferred with IQ-TREE [48] on 22,080 concatenated SNPs using the TVM+F+ASC+R2 substitution model [49]. Four clear populations are present, corresponding to ST58, ST60, ST1005, and ST93, with limited admixing between STs. Each color represents a population and each individual genome is displayed as a horizontal line subdivided into color blocks whose lengths represent the admixture proportions from K = 4 populations.

A concatenated SNP alignment from 169 INT2 genomes identified 7,960 core genome SNPs, out of 22,080 total SNPs, with a consistency index ≤0.5. The locus tags in K96243 with the highest number of homoplastic SNPs in INT2 genomes include: BPSS1007 (polyketide synthase), BPSS0409 (hypothetical protein), BPSS0306 (polyketide synthase), and BPSL1712 (non-ribosomal antibiotic-related protein synthase); the functions of these and many other regions containing homoplastic SNPS are largely unknown.

## Comparative genomics of representative STs

Sub-trees were generated for the four major STs (ST60, ST58, ST1005, ST93) (S1–S4 Figs, respectively). Although gene content variability within each ST was largely due to convergent gene loss, unique genes were also identified in a subset of strains; unique and lost genes are represented as a heatmap correlated to the phylogeny (S1–S4 Figs). A pan-genomics analysis determined that the core and pan-genome for each ST were largely similar in size (Table 1),

**Table 1. Coding region stats for each major INT2 sequence type.**

| Population | #genomes | Pan-genome size (#CDSs) | Core-genome size (#CDSs) | #Unique variants | Unique locus tags |
|---|---|---|---|---|---|
| 93 | 82 | 6185 | 4885 | 1 | AQ771_RS06540 |
| 58 | 16 | 5834 | 5604 | 1 | C1W81_11530 |
| 1005 | 34 | 5896 | 5674 | 2 | C1W93_05955,C1W93_24425 |
| 60 | 37 | 5963 | 5551 | 2 | C1W75_00010,C1W75_27835 |

suggesting minimal, yet observable gene differences within each ST. When compared to the global set of *B. pseudomallei* genomes, unique gene variants were identified within each of the four STs (Table 1); although homologs were observed for these unique regions in other *B. pseudomallei* genomes, each ST-specific sequence contained unique insertions/deletions that were not observed in these other genomes and, thus, represent genes that encode distinct proteins.

## Bayesian timing analysis

The 169 INT2 *B. pseudomallei* genomes representing the four populations identified by fastStructure were included in the Bayesian convergence analysis. The SNP matrix included reference positions present in >90% of reads and had a minimum coverage of 10X that spanned 90.1% of reference chromosome I in K96243 (NC_006350.1) and 75.1% of reference chromosome II (NC_006351.1).

Based on previous research [55], the estimated mutation rates for chromosome I and chromosome II differ slightly, $9.22E^{-7}$ vs $6.71E^{-7}$. As such, the time to most recent common ancestor (TMRCA) was calculated in the BEAST analysis separately for each chromosomal dataset using these previously reported mutation rates. For the 169 INT2 genomes, the mean TMRCA for chromosome I was estimated at 1258.5 years ago (95% highest posterior density (HPD), 1229.7, 1286.5), and the mean TMRCA for chromosome II was estimated at 791.0 years ago (95% HPD, 757.8, 823.1; Fig 4). The TMRCA for the ST1005 clade was estimated at 80.7 years ago (95% HPD, 72.6, 88.9) for chromosome I and 48.7 years ago (95% HPD, 40.1, 57.3) for chromosome II. The TMRCA for ST58 clade was estimated at 243.8 years ago (95% HPD, 229.6, 258.1) for chromosome I and 63.6 years ago (95% HPD, 53.2, 73.8) for chromosome II. The ST93 lineage TMRCA was estimated to be 95.4 years ago (95% HPD, 87.1, 104.0) for chromosome I and 74.5 years ago (95% HPD, 63.4, 85.0) for chromosome II. The ST60 clade TMRCA was estimated to be 250.3 (95% HPD, 238.9, 262.0) for chromosome I and 67.4 years ago (95% HPD, 58.4, 76.1) for chromosome II.

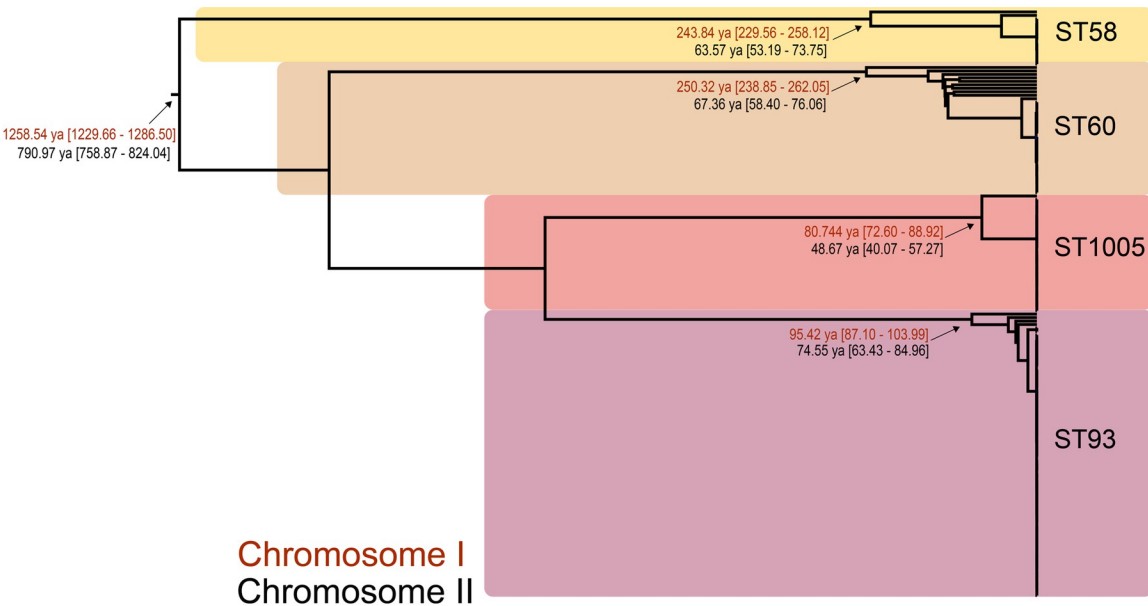

**Fig 4. Estimate of times to most recent common ancestor for the 169 INT2 *B. pseudomallei* genomes using BEAST [56].** Node dating ranges for both chromosomes (chromosome I in red, chromosome II in black) are shown on the Bayesian phylogeny. STs are indicated for each population.

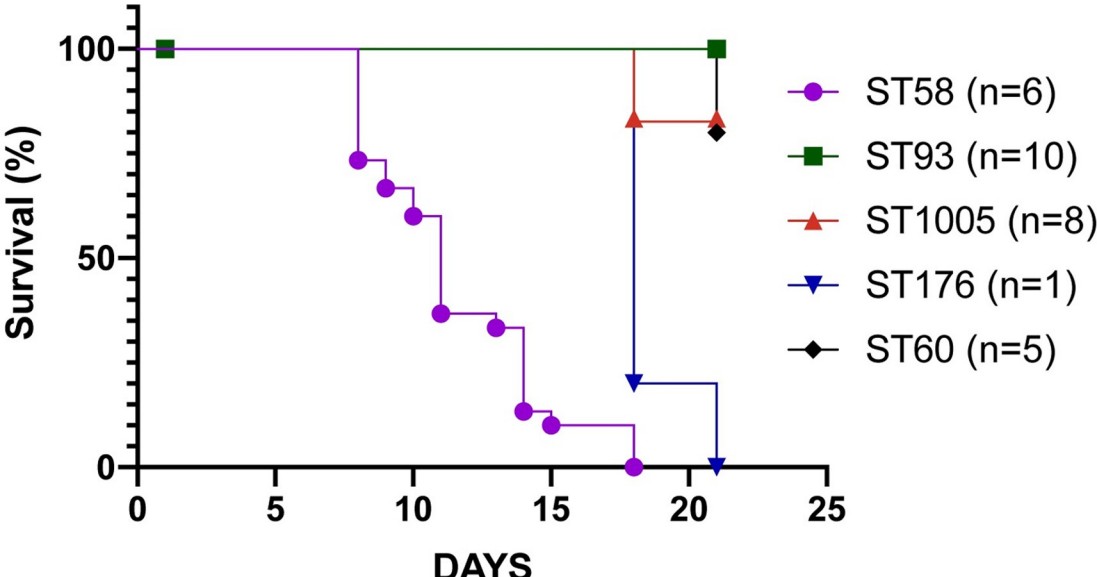

**Fig 5. A kill curve based on subcutaneous injection of 500 CFUs in 5 mice for each isolate.** The mortality curves were plotted in Prism. For each ST, all mice were pooled, and mortality curves plotted. Raw survival data per isolate is shown in S3 Table.

## Differential virulence

The virulence of 30 INT2 *B. pseudomallei* isolates assigned to five different sequence types was examined in a murine infection model (5 mice per strain); these included isolates from the four main STs (ST58, ST60, ST93, ST1005), as well as isolate Bp8873 from ST176 (S1 Table); ST176 is a single locus variant of ST60. In addition, two controls strains with known variable virulence in this model (NCTC13178, NTCT13179) were processed at the same dosage. The probability of survival was calculated from the pooled challenged mice per tested ST (Fig 5, raw data in S3 Table). A core genome phylogeny of 169 INT2 isolate genomes demonstrates the phylogenetic distribution of the 30 isolates utilized for virulence studies within this larger set (S5 Fig). Isolates were considered virulent if they killed all tested mice, attenuated if they killed no mice, and intermediate if they killed between 1 and 5 mice. None of the tested ST93 isolates (n = 10) killed any mice based on the subcutaneous inoculation pathway. In contrast, all tested ST58 isolates (n = 6) were highly virulent, killing all challenged mice. Interestingly, six ST1005 strains killed no mice, whereas 1 ST1005 strain (Bp8884) killed all mice challenged (Fig 5 and S3 Table). Four ST60 isolates did not kill any challenged mice, whereas one isolate (Bp8881) killed one mouse. The single ST176 isolate (Bp8873) was highly virulent, killing all challenged mice before the end of the 21-day experiment. The virulent control strain (NTCT13178) killed all mice at 500 CFU by day 5 and the less virulent control strain killed no mice over the 21-day experiment.

## Genome wide association studies

To identify genotypic differences that may explain the virulent phenotype, the program pyseer was run on a set of all 54-mers from Illumina reads from each of the 30 INT2 isolates used for virulence studies. Pyseer identified 31 virulence-associated 54-mers, all of which mapped to a single annotated gene, *mgtA* (C1W34_19470). However, all of these Kmers had "bad chisq" values signifying possible spurious associations; a "bad chisq" value is reported by pyseer and no additional information is available. These 31 associated Kmers were conserved in 7 of 8

virulent genomes and 8 of 20 attenuated genomes. However, a manual inspection of an alignment of this gene demonstrated that SNPs were highly correlated with population structure, with the exception of a single SNP found in all ST58 and ST60 genomes (including ST176), most likely due to homologous recombination.

In addition to pyseer, we also used a machine learning (ML) Boruta model to identify CDSs potentially associated with virulence. ML has been previously used to accurately predict phenotype from genotype [72,73]. Although ML methods used in this study have not been previously used for bacterial GWAS, the Boruta method has been established by statisticians and human GWAS researchers as an efficient and specific feature selection algorithm [66,74]. By iterating over the random seed of Boruta, we were able to identify 45 associated CDSs that were consistently selected as associated with virulence, although only 11 of them (S6 Fig) were more conserved in virulent genomes. Some of these regions demonstrate a population structure effect, whereas others represent a polyphyletic distribution, most likely due to recombination.

## Pan-genomics of animal-challenged genomes

A pan-genomics analysis of the 30 INT2 isolates utilized in the virulence study demonstrated a core-genome size of 5,369 CDSs and a pan-genome size of 6,338 CDSs. To identify potential differences in the pan-genome between virulent and attenuated strains, an LS-BSR analysis was performed, which failed to identify any genomic regions that fully differentiated between these two phenotypes. However, at a BSR threshold of 0.75, there were two CDSs (C1W87_17265; hypothetical protein, AQ770_28965; non-ribosomal peptide synthetase) that were highly conserved in the genomes of 7 of 8 virulent isolates and less conserved in the genomes of all 20 attenuated isolates (S6 Fig); these same two regions were also identified by the machine learning method. Of the seven virulent isolates possessing these regions, six were assigned to ST58 and one to ST176 (Bp8873). ST176 is a single locus variant of ST60 and none of the ST60 isolates exhibited the virulent phenotype. The polyphyletic distribution of these 2 genes (S6 Fig) suggests that these regions were a product of recombination and could be the subject of additional investigation.

For ST1005, six of the isolates were attenuated, one demonstrated intermediate virulence, and one of the isolates was virulent. A comparative analysis failed to identify CDSs that fully differentiated between phenotypes within this ST. A Kmer analysis identified 31 Kmers that were associated with a single locus (C1W93_20275) that was enriched in the genome from the virulent isolate; this was the same locus that was identified by pyseer on the complete set of animal-challenged genomes.

A screen of previously characterized virulence factors across the genomes of the 30 INT2 isolates included in the virulence study identified that Hcp1, which has previously been linked with virulence in *B. pseudomallei* [23], was associated with the virulent phenotype; this region was also identified with the ML method. All 8 of the virulent strains had the same version of Hcp1 found in K96243 (BPSS1498). However, 14 of 20 attenuated isolates, plus one intermediate virulence isolate (Bp8881), contain a variant of Hcp1 (C1W89_RS19320) with significant amino acid differences to BPSS1498 (S7 Fig). This demonstrates that this Hcp1 homolog has a different peptide composition and may have a different protein activity associated with attenuated virulence. A screen of the nucleotide sequences of Hcp1 against all ST93 genomes in the GenBank assembly database demonstrated that both alleles are present in INT2 ST93 genomes (S8 Fig), demonstrating recombination of this region; INT2 genome Bp1782, in particular, has an annotated Hcp1 that is highly divergent at both the nucleotide and amino acid level (identical to another annotated Hpc1 gene (BBU_3899)), suggesting that it may have altered function.

Six attenuated strains from other STs (Bp8894, Bp8892, Bp8871, Bp8890, Bp8886, Bp1927) had the K96243 version of Hcp1, suggesting that a different mechanism of attenuation exists in those strains. Interestingly, Panaroo grouped both Hcp1 variants (BPSS1498, C1W89_RS19320) together into a single gene and the binary presence/absence results demonstrate that all animal-challenged genomes contain the K96243 Hcp1 allele; using Panaroo alone on this dataset would not have identified the ST93 Hcp1 differences.

For the remaining 6 attenuated ST1005 isolates with the K96243 Hcp1 allele (Bp8892, Bp8886, Bp8890, Bp8894, Bp8871, Bp1927), a comparison was performed with all virulent INT2 isolate genomes (n = 8) using a set of previously characterized virulence genes in *B. pseudomallei*. The results demonstrate that a cluster of genes (BPSS0417-BPSS0428) associated with virulence was conserved in 7 of 8 virulent isolates from the other STs (BSR value ≥0.8) and was missing from all ST1005 isolates (6 attenuated and 1 virulent isolates); these genes are all part of a polysaccharide biosynthetic operon that has been demonstrated to react strongly with patient sera [75].

## Discussion

In this study, we explored the genomic diversity and differential virulence of *B. pseudomallei* within a single soil sample collected from a rice paddy in the Ubon Ratchathani Province of Thailand. MLST analysis identified 7 unique sequence types (STs) within a single soil sample, highlighting genotypic diversity and prompting the investigation into whole genome comparisons and *in vivo* animal challenge.

Comparative genomics identified variably conserved coding region sequence (CDS) differences between STs as well as within STs. Genomes from a single ST were not monomorphic, demonstrating within-ST diversity from a single soil sample. The within-ST diversity in these isolates was only apparent by sequencing multiple bacterial colonies from the same isolation plate. If only a single colony was picked from this soil sample during propagation, the comprehensive sample diversity described here would have been missed. If a plate sweep had been sequenced, a ST mixture would have been recovered and would have complicated all downstream population analyses. As demonstrated by this study, an analysis of individual colonies was the best option to identify and understand the sample diversity and should be considered for comprehensive environmental sampling of *B. pseudomallei*.

An investigation into population structure identified 4 distinct populations (Fig 3), with demonstrated mixing between a small number of genomes. Horizontal gene transfer and recombination have been demonstrated to help drive the diversification in *B. pseudomallei* [19], and the observation of recombination in genomes from a single sample is not surprising given their proximity and related genomic background. The CDSs that demonstrated the highest levels of homoplasy were associated with non-ribosomal protein synthetases; the function of many of these regions is not known but suggests potential mechanisms of environmental adaptation. Other recombination events were observed in virulence-associated genes (S6 Fig), including in Hcp1, a gene associated with secretion of type six effectors and disease in mammals. This result suggests that recombination could drive virulence in environmental *B. pseudomallei* strains.

The subcutaneous (SC) BALB/c model was used to identify variable virulence as it simulates percutaneous infection [7], has published $LD_{50}$ values [60], and has been used in other virulence studies [76,77]. We did not perform histology in this study and it is unclear if surviving mice were colonized by *B. pseudomallei*. The results do demonstrate that different strains kill mice at a similar dosage, which may correlate to variable disease presentation in human infections. Variable virulence was observed among isolates from INT2 and was largely segregated

by ST. For example, ST93 was found to be highly attenuated in the mouse model based on the dosage and route of exposure. A screen of public genomes in GenBank identified 8 ST93 genomes in addition to those sequenced in this study; 6 of these additional ST93 genomes were isolated from the environment whereas the other 2 were reported to be isolated from humans, although details on isolation source were missing. In general, the results of this study demonstrate that ST alone is insufficient to predict virulence in *B. pseudomallei*. A recent study identified genomic differences between clinical and environmental isolates [16], suggesting that some environmental isolates may not contain genomic elements required for virulence in humans.

The animal study prompted analysis into genes associated with the virulent phenotype. A screen of genes previously associated with virulence identified that all attenuated ST93 genomes have an alternative allele for Hcp1, an important virulence factor in *B. pseudomallei*. Of external ST93 genome assemblies, 3 have the K96243 version of Hcp1, while 5 have the alternate allele; one of the external genomes with the alternate Hcp1 allele was associated with humans (SAMN04208633), although details of the isolation source are not available in the public record. Previous results demonstrated that even single amino acid differences can result in different phenotypes against a similar genomic background [78]. Although most ST93 genomes sequenced in this study (80/82) have the alternative Hcp1 allele, two genomes have an identical allele to Hcp1 in *B. pseudomallei* K96243 (S8 Fig); the presence of both alleles within a single soil sample is most likely explained by homologous recombination. Testing these naturally occurring, related strains in a murine melioidosis model will demonstrate if this one allele difference is responsible for the differential virulence observed in this sequence type; if virulence is observed in the wild strain, allele replacement will confirm that Hcp1 confers the virulent phenotype. For other genomes associated with virulence, machine learning methods identified additional regions that were not strictly associated with population structure, suggesting that the apparent recombination may be associated with the observed virulence phenotype in the murine model, but additional experimentation is required to validate these results and benchmark the Boruta ML method. The lack of a single region that explains the virulence phenotype suggests that either multiple independent mechanisms are associated with virulence or that complex interactions between mechanisms result in the virulent phenotype.

Our findings indicate that environmental exposure to *B. pseudomallei*, especially inoculation via contact with contaminated soil and water, but possibly also via inhalation and ingestion, likely means exposure to a mixed community of *B. pseudomallei* strains. Attenuated strains are likely to be quickly cleared by the host immune system, although the virulent strains may cause disease in sensitive patients. A recent longitudinal study of melioidosis in a single patient demonstrated the selection for specific genotypes with significantly reduced virulence in chronic infections [78], suggesting that either the immune system can select for strains from the environment that cause attenuated, but chronic infection, or that strains that evade the immune system persist in the human host.

In addition to being a significant public health threat, *B. pseudomallei* is also a biothreat agent and our results have implications for studies of the pathogen in this context. The diversity of INT2 isolate genomes, measured by total pairwise SNPs as well as pan-genome size, is greater than the known, global diversity of other biothreat pathogens, including *Bacillus anthracis* [79], *Yersinia pestis* [80], and *Francisella tularensis* [81]. This diversity complicates sample matching [82], suggesting that matching disease to a specific strain using shared mutations may need to be supplemented by overlap in the accessory genome. A recent study identified unique genomic islands in isolates from Puerto Rico [10], suggesting that whole genome comparisons are needed for robust sample matching.

This study presents a focused survey into the evolution, diversity, and virulence of *B. pseudomallei* isolates from a single soil sample. Based on the projected distribution of *B. pseudomallei* throughout the world [83], there likely exists an underexplored diversity of strains that have variable risk to humans. This study provides a framework for studying this diversity and identifies genomic targets that may focus additional studies to better understand the virulence of *B. pseudomallei*.

## Supporting information

**S1 Fig. A maximum-likelihood phylogeny of ST93 genomes inferred from a concatenation of 15,249 SNPs with IQ-TREE [48] using the TVM+F+ASC+R3 substitution model [49].** The phylogeny was annotated with coding regions that show variable distribution, based on an analysis with LS-BSR [38]. The phylogeny and heatmap were visualized with the interactive tree of life [41] and rooted with *B. pseudomallei* K96243 [44] as it represents an outgroup genome from Thailand.
(TIF)

**S2 Fig. A maximum-likelihood phylogeny of ST58 genomes inferred from a concatenation of 27,500 SNPs with IQ-TREE [48] using the GTR+F+ASC+R2 substitution model [49].** The phylogeny was annotated with coding regions that show variable distribution, based on an analysis with LS-BSR [38]. The phylogeny and heatmap were visualized with the interactive tree of life [41] and rooted with *B. pseudomallei* K96243 [44] as it represents an outgroup genome from Thailand.
(TIF)

**S3 Fig. A maximum-likelihood phylogeny of ST60 genomes inferred from a concatenation of 23,735 SNPs with IQ-TREE [48] using the TVM+F+ASC+R4 substitution model [49].** The phylogeny was annotated with coding regions that show variable distribution, based on an analysis with LS-BSR [38]. The phylogeny and heatmap were visualized with the interactive tree of life [41] and rooted with *B. pseudomallei* K96243 [44] as it represents an outgroup genome from Thailand.
(TIF)

**S4 Fig. A maximum-likelihood phylogeny of ST1005 genomes inferred from a concatenation of 20,923 SNPs with IQ-TREE [48].** The phylogeny was annotated with coding regions that show variable distribution, based on an analysis with LS-BSR [38] using the TVMe+ASC +R3 substitution model [49]. The phylogeny and heatmap were visualized with the interactive tree of life [41] and rooted with *B. pseudomallei* K96243 [44] as it represents an outgroup genome from Thailand.
(TIF)

**S5 Fig. A maximum-likelihood phylogeny of all sequenced INT2 genomes (n = 169) inferred from a concatenation of 22,080 SNPs with IQ-TREE [48] using the TVM+F+ASC +R2 substitution model [49].** Animal-passed genomes are colored by attenuated, intermediate, or virulent phenotypes. The phylogeny is rooted by *B. pseudomallei* K96243 [44] as it represents an outgroup genome from Thailand.
(TIF)

**S6 Fig. A maximum-likelihood phylogeny of genomes from animal-challenged INT2 isolates inferred from a concatenation of 32,821 SNPs with IQ-TREE [48] using the TVM+F +ASC substitution model [49].** Genomes are colored by the virulence outcome in an animal challenge model. Each genome was screened with LS-BSR [38] using genes identified through

machine learning methods. The phylogeny and heatmap were visualized with the interactive tree of life [41] and rooted with *B. pseudomallei* K96243 [44] as it represents an outgroup genome from Thailand.
(TIF)

**S7 Fig. An alignment of protein sequences from Hcp1 (BPSS1498) and a novel allele variant seen in ST93 genomes (C1W89_RS19320).** Blue boxes surround amino acid differences in the alignment, which was visualized with Jalview [65].
(TIF)

**S8 Fig. A maximum-likelihood phylogeny of ST93 genomes sequenced in this study as well as GenBank external genomes inferred from a concatenation of 37,746 SNPs with IQ-TREE [48] using the TVM+F+ASC substitution model [49].** The phylogeny was rooted with *B. pseudomallei* K96243 [44]. Each genome was screened with LS-BSR [38] using two Hcp1 variants. The phylogeny and heatmap were visualized with the interactive tree of life [41] and rooted with *B. pseudomallei* K96243 [44] as it represents an outgroup genome from Thailand.
(TIF)

**S1 Table. Accession information for genomes sequenced in this study.**
(XLSX)

**S2 Table. Accession information for a global set of *B. pseudomallei* genomes.**
(XLSX)

**S3 Table. Number of mice remaining after each day of the mouse challenge experiment.**
(XLSX)

## Acknowledgments

The authors thank Nathan Stone for generating the Thailand map in Fig 1.

## Author Contributions

**Conceptualization:** Richard A. Bowen, Vanaporn Wuthiekanun, Sharon Peacock, Apichai Tuanyok, David M. Wagner.

**Data curation:** Chandler Roe, Adam J. Vazquez, Jason W. Sahl.

**Formal analysis:** Chandler Roe, Adam J. Vazquez, Paul D. Phillips, Chris J. Allender, Richard A. Bowen, Roxanne D. Nottingham, Adina Doyle, Gumphol Wongsuwan, Jason W. Sahl.

**Investigation:** Paul D. Phillips, Jason W. Sahl.

**Methodology:** Chris J. Allender, Richard A. Bowen.

**Project administration:** Vanaporn Wuthiekanun, Direk Limmathurotsakul, Sharon Peacock, Paul Keim, Apichai Tuanyok, David M. Wagner, Jason W. Sahl.

**Software:** Jason W. Sahl.

**Supervision:** David M. Wagner.

**Writing – original draft:** Chandler Roe, David M. Wagner, Jason W. Sahl.

**Writing – review & editing:** Jason W. Sahl.

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
