## [Decision Letter · Decision Letter 0]

20 Oct 2021

Dear Dr. Sahl,

Thank you very much for submitting your manuscript "Multiple phylogenetically-diverse, differentially-virulent Burkholderia pseudomallei isolated from a single soil sample collected in Thailand" for consideration at PLOS Neglected Tropical Diseases. As with all papers reviewed by the journal, your manuscript was reviewed by members of the editorial board and by several independent reviewers. In light of the reviews (below this email), we would like to invite the resubmission of a significantly-revised version that takes into account the reviewers' comments. 

All reviewers think the study is potentially interesting and can contribute to the field. However, all have point out problems with some interpretation of the data, that the authors have to address. Particularly, reviewer 2 highlights major flaws in the animal model chosen for virulence. I would agree that the authors have to provide evidence why this is a valid mouse model that approximates virulence either through well defined virulent vs non-virulent isolates performed in this model, to have a "calibration" standard. Otherwise, the authors should consider using more standard and well established models of virulence either through ip, iv, intranasal or intratracheal routes.

We cannot make any decision about publication until we have seen the revised manuscript and your response to the reviewers' comments. Your revised manuscript is also likely to be sent to reviewers for further evaluation.

Sincerely,

Yunn-Hwen Gan

Associate Editor

Elsio Wunder Jr

Deputy Editor

All reviewers think the study is potentially interesting and can contribute to the field. However, all have point out problems with some interpretation of the data, that the authors have to address. Particularly, reviewer 2 highlights major flaws in the animal model chosen for virulence. I would agree that the authors have to provide evidence why this is a valid mouse model that approximates virulence either through well defined virulent vs non-virulent isolates performed in this model, to have a "calibration" standard. Otherwise, the authors should consider using more standard and well established models of virulence either through ip, iv, intranasal or intratracheal routes.

Reviewer's Responses to Questions

**Key Review Criteria Required for Acceptance?**

**Methods**

-Are the objectives of the study clearly articulated with a clear testable hypothesis stated?

-Is the study design appropriate to address the stated objectives?

-Is the population clearly described and appropriate for the hypothesis being tested?

-Is the sample size sufficient to ensure adequate power to address the hypothesis being tested?

-Were correct statistical analysis used to support conclusions?

-Are there concerns about ethical or regulatory requirements being met?

Reviewer #1: Methods are sound and comprehensive

Reviewer #2: (No Response)

Reviewer #3: The study clearly describes its objective of investigating the genomic diversity and differential virulence of B.pseudomallei isolates obtained from a single soil sample. 

The approaches taken to investigate the stated goals were appropriate. To investigate genomic diversity, the authors subjected B. pseudomallei isolates from the soil sample to genome sequencing, in silico genotyping and phylogenetic analysis. To investigate the virulence of the strains, a murine infection model via subcutaneous injection of B. pseudomallei was adopted. However, only 30 out of the 169 B. pseudomallei isolates sequenced were investigated for their virulence potential in the in vivo animal challenge. There is no explanation on why these 30 strains were chosen. 

There are no obvious identifiable ethical concerns. The authors had declared the use of IACUC- approved protocols, appropriate risk group 3 biosafety practices and compliance with the Guide for the Care and Use of Laboratory Animals of NIH. The genomic data of all sequenced strains are available on GenBank.

**Results**

-Does the analysis presented match the analysis plan?

-Are the results clearly and completely presented?

-Are the figures (Tables, Images) of sufficient quality for clarity?

Reviewer #1: Results are clear and thorough

Reviewer #2: (No Response)

Reviewer #3: The analysis performed matched the objectives of the study. However, clarity of the results and terms used can be improved.

The authors should have included explanations of why certain strains were chosen for rooting of the phylogenetic trees. For example, it should have been explained that B. pseudomallei strain MSHR0668 is the most ancestral B. pseudomallei strain identified in a previous phylogenetic study. It is not apparent what is the rationale for using K96243 for rooting the trees in Figures S1 to S4. 

The term “unique genomic regions” (Line 345-346) is confusing since homologs of these genomic regions are found in other B. pseudomallei strains and not only in the STs examined. “Unique insertions/deletions” (Line 348) suggest that each ST has distinct variants but not a “unique genomic region”.

The consolidated survival curve (Fig 5) provides only a general overview of the virulence potentials of each ST. While the authors did provide supplementary data on the number of mice surviving throughout their in vivo mouse challenge experiment (Table S3), the data presented is raw. The strains in the table should be organized according to ST for easier viewing. The survival curves for all the strains should also be plotted and presented for readers to quickly identify which strains are more virulent/attenuated or behaved exceptionally in comparison to the other strains within the same ST. 

The terms “virulent”, “attenuated” and “intermediate”; which were used to describe strains that killed all mice, no mice and some mice respectively; were only defined in the Materials and Methods section. The authors should also clearly define these terms within the main text (Under the section on “Differential virulence” which examines virulence of the different strains in the in vivo mouse model) for clarity in the subsequent sections. 

The authors mentioned “14 out of the 20 attenuated isolates… contain a variant of Hcp1” (in Line 433) and “Five attenuated strains from other STs had the K96243 version of Hcp1” (in Line 442). These descriptions are vague, and it is unclear which isolates have the C1W89_RS19320 Hcp1 variant or K96243 version of BPSS1498. A table should be included to clarify this.

A more thorough explanation of the datasets and the concluding statements would also enhance readability. For instance, in lines 340 to 343, the authors stated that gene variability in each ST “was largely due to convergent gene loss” without explaining how the data shows that. Is the loss/gain of genes relative to a more ancestral strain in each ST (i.e. in ST93, Bp1740) from the INT2 soil sample? This should be explicitly stated.

**Conclusions**

-Are the conclusions supported by the data presented?

-Are the limitations of analysis clearly described?

-Do the authors discuss how these data can be helpful to advance our understanding of the topic under study?

-Is public health relevance addressed?

Reviewer #1: Conclusions are mostly well supported. Detail provided in comments to the authors

Reviewer #2: (No Response)

Reviewer #3: Key discussion of the data, implications of the study, and conclusions of the results were generally appropriate, with some points requiring clarification. 

In the abstract, the authors wrote “Five different sequence types were identified” (Lines 58-59). All the data described and shown indicates that there were 7 sequence types observed in the one soil sample. 

“The core genome phylogeny demonstrated that genomes from this study shared a recent common ancestor (Figure 2), suggesting a common geographic origin.” (Lines 301-302). Does this really suggest a common geographic origin? In the phylogenetic tree (Fig 2), there were GenBank strains that branched out from the INT2 common ancestor. Are those strains also isolated from the same region in Thailand? 

In Lines 343-345, the authors mentioned that “A pan-genomics analysis determined that the core and pan-genome for each ST were largely similar in size (Table 1), suggesting minimal, yet observable gene differences within each ST.” This statement needs to be revised. Size does not necessarily correlate to minimal gene differences. The integration of new genes coupled with the loss of others could also result in minimal changes in size.

The authors found that virulence was largely segregated by the STs (Fig 5 and line 485-486). This suggests that ST may be able to provide a general indication of virulence, although it may not be the most accurate, since certain strains within the ST may display exceptions. The author’s statement of ST being “a poor predictor of virulence” (Line 493) will therefore need to be further justified. Discussions on ST93 (in lines 486-492) are not able to support the author’s view of ST being a poor virulence predictor. Further discussion on the other STs - ST58 (which was virulent), ST1005 and ST60 (largely attenuated with several exceptions) – should be made to justify the author’s stand.

**Editorial and Data Presentation Modifications?**

Reviewer #1: (No Response)

Reviewer #2: (No Response)

Reviewer #3: The recommended modifications have been described in all the other sections.

**Summary and General Comments**

Reviewer #1: The authors present an interesting study on the genomic diversity and virulence potential of the population of B. pseudomallei in a single soil sample from Thailand. I enjoyed reading the manuscript and have only minor comments to the authors.

Minor points

Line 36. Suggest “all identified STs” rather than “all STs” for clarity.

Line 36. You can use the abbreviation STs here. 

Line 42-44: Suggest re-wording for clarity. Is virulence being stratified by ST suggestive of convergent evolution or is one virulent isolate and six attenuated isolates belonging to the same ST suggestive of convergent evolution? Differing virulence within the same ST would be the opposite of convergent evolution.

Line 83. The study by McRobb et al., 2014 (https://doi.org/10.1128/AEM.00128-14) is important to mention here to capture the Australian context. This study used MLST to look at the population of ~170 Australian B. pseudomallei from the environment across the Top End region. 

Line 357. The mutation rate for chromosome two has been listed here as 9.71e-7 but listed previously as 6.71e-7. Please check for consistency.

Line 398. What exactly is a “bad chisq” value? Are you saying that they haven’t reached statistical significance? More explanation is needed here. What cut-offs were used to establish a bad or good chisq test? Was the data corrected for multiple testing?

Line 415-416. The locus names are not particularly useful here. Is there a homolog in K96243 that could be used to define these two loci?

Line 471-2. As a comment. Yes, the ST mixture would have complicated all downstream analyses, but it may have captured additional diversity not observed with the colony picking methods. We need improved methodology to sample complete diversity and be able to analyse this complex milieux of data.

Line 483. A reference around hcp function would be good here to back-up the statement about disease in mammals.

Line 485-6. Suggest using the word “sequence type” rather than “genotype”.

Line 490-492. “This suggests that either this ST exhibits low virulence in humans and is therefore not showing up in clinical surveillance or has not been observed in clinical samples due to insufficient sampling.” This statement is unsupported by the data. B. pseudomallei is not considered a commensal organism; therefore, according to the previous line, ST93 has been isolated from two human cases of melioidosis. The data presented suggests that ST93 may not display virulence in the murine model but can still infect humans and cause melioidosis. There are multiple cases of a single ST being isolated from both environmental and clinical sources so finding slightly more ST93 in the environment vs clinical studies does not support the argument of this being an attenuated ST in humans.

Line 492. “In general, the results of this study demonstrate that ST alone is a poor predictor of virulence in B. pseudomallei.” This is in perfect agreement with a much larger study that looked at the distribution of STs in a large clinical cohort and in the environment. They saw that there was no difference in the clinical vs environmental ST distribution across the entire Top End of Australia (McRobb et al., 2014 - https://doi.org/10.1128/AEM.00128-14).

Line 501. B. pseudomallei is not considered a commensal. Granted that this case may have been a mild case of melioidosis but an isolate from a human is from a case of melioidosis. 

Line 505-507. This experiment may help determine the role of Hcp1, however any additional variation in the genomes would still have to be taken into account. A better approach would be to use isogenic strains with a genetically modified hcp1 allele to ensure that no other differences are present. As stated, the experiment has been oversimplified.

Line 510-513. Worth restating that this is a murine model of virulence that has been investigated.

Line 515. Suggest removing “and water”. The study did not investigate population diversity in water and makes no claim as to the diversity present. 

Line 520-1 As a comment. “immune system can select for strains from the environment that cause attenuated, but chronic infection” or the immune system can eliminate some strains but only those that can evade the immune system survive and cause chronic infection. In addition to P314, the same virulence attenuation has been observed in multiple cases of B. pseudomallei infection in cystic fibrosis patients (10.1128/mBio.00356-17). 

Line 526-528. The addition of indels has also helped differentiate strains in cases where SNPs fail to provide enough resolution. This approach has been well established for B. pseudomallei too (e.g. 10.1099/mgen.0.000067)

Reviewer #2: This manuscript describes a range of studies conducted on B. pseudomallei isolates obtained from a grid of soil samples collected in 2007 from a rice field located in a region of Thailand endemic for the disease melioidosis. The data presented are interesting in that they are suggestive of variability of B. pseudomallei sequences within a small geographic area. However, the samples are obtained from soil samples collected 14 years ago, the animal model experiments lack rigour and some of the bioinformatic analysis lacks clarity in its approach and is questionable in its interpretation. These weaknesses in the study unfortunately undermine the conclusions of the authors who suggest there are significant levels of differential virulence that could be ascribed to genomic differences.

Specific areas of concern/improvement are listed below.

1. The introduction suggests that the most probable route of infection occurs through bacterial contamination of wounds. The limited literature cited to support this statement lack specific evidence for this and the literature cited is not a primary reference. The statement also ignores the presentation of cutaneous melioidosis, which can also be a self-limiting infection. Whilst infection via wounds is considered a viable route of infection, the authors must cite more published data to indicate that this is the most probable route of infection compared to ingestion and inhalation, assuming that is true.

2. Another point of concern is the source of the material. It appears that the samples are recently isolated from soil obtained in 2007. It is not clear why aged samples were used and how these samples were stored over such a long period of time. Could sample age and the nature of the storage (along with other microbes that are present in these samples) have influenced the sequencing diversity reported via recombination or other mechanisms that result in genetic modifications?

3. The animal model used (sub-cutaneous inoculation with 5x10^2) is not well established for assessing virulence. There is no literature cited in the manuscript that indicates studies have been done to establish lethal doses of B. pseudomallei in this murine model as compared to other murine models that use i.p. or i.n. as a route of infection. Developing this s.c. model should include prototypical bacterial strains including the K96243 and the l attenuated strain B. thailandensis E264. MOIs could differ significantly between the strains studied, which in turn does not mean that any one strain is less lethal than another – particularly given the very low challenge dose. The complete lack of any pathology – even bacterial counts on vital organs is a considerable weakness if not a fatal flaw that is also standard practice in assessing virulence in melioidosis infection models. 

4. The genomic assembly, genotyping, comparative pan-genomics, and SNP identification and polymorphism analysis are well-established methods are data presented appear to be appropriately analysed with acceptable statistics. However, feature identification using the ML was not particularly well-justified or convincing. Benchmark studies are needed here to demonstrate that such an approach is viable – perhaps using genomes from related type strains of B. pseudomallei, B. mallei and B. thailandensis – and/or the collection of B. pseudomallei strains already deposited in GenBank/NCBI. 

5. The identification of Hcp1 as displaying significant amino acid differences that in turn may suggest it could have a different function is not supported by the data. There is no citation to indicate how they conducted their pairwise alignment nor statistics to indicate that these differences are truly significant. A standard pairwise comparison using the standard Needle-Wunsch algorithm gives identity and similarity percentages of approximately 82% and 89%, respectively - and a slightly modified alignment as well. These percentages are suggestive of conserved structure and function and are nowhere near “twilight” values (e.g. <40%) that are typically indicative of an different function and possibly significant structural differences. Similarly, the identification the loss of a gene cluster involved in polysaccharide synthesis may be significant but would need some minimal demonstration comparisons with patient sera and even API data that could support the presumption this might influence reduced virulence of the ST1005 isolate. The importance of this strain in human infections is speculative at this point and it is possible that more virulent strains would simple out-compete this strain in the more likely scenario of a poly-microbial inoculation.

Hence, while the authors do rightly point out that there likely exists an under-explored diversity of strains that could pose risks to humans, a considerable amount of data presented in this manuscript are too preliminary or unconvincing to adequately support this statement. It is questionable whether these strains demonstrate significant levels of differential virulence/attenuation in this study given lack of rigour in the animal model studies. The authors might consider revising this manuscript to focus on the genomic analysis, comparing their sequences with existing sequences already available in the public domain. Given that a protocol now exists for isolation and transfer of materials it may also be worth considering obtaining fresh (as opposed to aged) soil samples as another means to support the sequence diversity findings reported in this manuscript.

Reviewer #3: In this study, the authors did a systematic and comprehensive analysis of the genomic diversity and virulence potentials of B. pseudomallei strains in a single soil sample . They discovered that B. pseudomallei isolates display high genotypic diversity, where 4 distinct populations, admixing between STs and within-ST diversity was observed in the single soil sample. Amongst the 169 B. pseudomallei strains sequenced, the authors selected 30 isolates for further investigations on their virulence potentials in an in vivo murine model. Virulence of the strains were largely segregated according to the STs (i.e. all ST93 strains were attenuated, all ST58 strains highly virulent, ST60 and ST1005 were largely attenuated) with several exceptions. Subsequent GWAS and pan genomics analysis of the 30 isolates utilized in the virulence study revealed 11 CDS that were more conserved in virulent genomes, including Hcp1, a known virulence factor. Hcp1 associated with the virulent phenotype, where all virulent strains possess the Hcp1 version of BPSS1498 and 14 out of 20 attenuated strains has the C1W89_RS19320 Hcp1 variant. The high genomic diversity identified has implications on environmental sampling strategies that examines the risk to humans and virulence potential of B. pseudomallei strains. Further insights on which genes could be potentially contributing to the bacteria’s fitness in in vivo infections or virulent phenotypes have also been gained from the identification of conserved genes in more virulent genomes. However, there are some concerns in terms of the interpretation of results and clarity of methods.

PLOS authors have the option to publish the peer review history of their article (what does this mean?). If published, this will include your full peer review and any attached files.

Reviewer #1: No

Reviewer #2: No

Reviewer #3: No
---

## [Decision Letter · Decision Letter 1]

17 Dec 2021

Dear Dr. Sahl,

Thank you very much for submitting your manuscript "Multiple phylogenetically-diverse, differentially-virulent Burkholderia pseudomallei isolated from a single soil sample collected in Thailand" for consideration at PLOS Neglected Tropical Diseases. As with all papers reviewed by the journal, your manuscript was reviewed by members of the editorial board and by several independent reviewers. In light of the reviews (below this email), we would like to invite the resubmission of a significantly-revised version that takes into account the reviewers' comments. 

Dear authors,

The reviewer has given detailed constructive comments pertaining to revision 1 and there are two aspects that require further revision. Although it is classified as major revision, the 2 points raised pertain to the interpretation and the conclusion of the findings which require the authors to acknowledge limitations and further discussion with inclusion of relevant references. I agree with the reviewer's recommendations and would ask the authors to consider the following.

1. The study did not undertake a proper virulence comparison either with surrogate models or further characterization of the mouse infection organ loads, LD50 etc. Therefore, discussion of limitations of this approach is warranted and the virulence comparison of the strains interpreted in this light.

2. More description of ML algorithms and justification, with limitations explained. It is indeed highly speculative that Hcp sequence variants result in altered protein function and virulence as Hcp is one of the highest conserved protein across species in T6SS secretion and function. Unless the authors have preliminary data to show that T6SS secretion is altered, this statement is very problematic. More analyses as suggested by reviewer should be considered.

We cannot make any decision about publication until we have seen the revised manuscript and your response to the reviewers' comments. Your revised manuscript is also likely to be sent to reviewers for further evaluation.

Sincerely,

Yunn-Hwen Gan

Associate Editor

Elsio Wunder Jr

Deputy Editor

Dear authors,

The reviewer has given detailed constructive comments pertaining to revision 1 and there are two aspects that require further revision. Although it is classified as major revision, the 2 points raised pertain to the interpretation and the conclusion of the findings which require the authors to acknowledge limitations and further discussion with inclusion of relevant references. I agree with the reviewer's recommendations and would ask the authors to consider the following.

1. The study did not undertake a proper virulence comparison either with surrogate models or further characterization of the mouse infection organ loads, LD50 etc. Therefore, discussion of limitations of this approach is warranted and the virulence comparison of the strains interpreted in this light.

2. More description of ML algorithms and justification, with limitations explained. It is indeed highly speculative that Hcp sequence variants result in altered protein function and virulence as Hcp is one of the highest conserved protein across species in T6SS secretion and function. Unless the authors have preliminary data to show that T6SS secretion is altered, this statement is very problematic. More analyses as suggested by reviewer should be considered.

Reviewer's Responses to Questions

**Key Review Criteria Required for Acceptance?**

**Methods**

-Are the objectives of the study clearly articulated with a clear testable hypothesis stated?

-Is the study design appropriate to address the stated objectives?

-Is the population clearly described and appropriate for the hypothesis being tested?

-Is the sample size sufficient to ensure adequate power to address the hypothesis being tested?

-Were correct statistical analysis used to support conclusions?

-Are there concerns about ethical or regulatory requirements being met?

Reviewer #2: Please see attached review

**Results**

-Does the analysis presented match the analysis plan?

-Are the results clearly and completely presented?

-Are the figures (Tables, Images) of sufficient quality for clarity?

Reviewer #2: Please see attached review

Figures and tables are clear

**Conclusions**

-Are the conclusions supported by the data presented?

-Are the limitations of analysis clearly described?

-Do the authors discuss how these data can be helpful to advance our understanding of the topic under study?

-Is public health relevance addressed?

Reviewer #2: Please see attached review

**Editorial and Data Presentation Modifications?**

Reviewer #2: (No Response)

**Summary and General Comments**

Reviewer #2: Please see attached review

PLOS authors have the option to publish the peer review history of their article (what does this mean?). If published, this will include your full peer review and any attached files.

Reviewer #2: No
---

## [Editor Report · Decision Letter 2]

14 Jan 2022

Dear Dr. Sahl,

We are pleased to inform you that your manuscript 'Multiple phylogenetically-diverse, differentially-virulent Burkholderia pseudomallei isolated from a single soil sample collected in Thailand' has been provisionally accepted for publication in PLOS Neglected Tropical Diseases.

Best regards,

Yunn-Hwen Gan

Associate Editor

Elsio Wunder Jr

Deputy Editor

---

## [Editor Report · Acceptance letter]

7 Feb 2022

Dear Dr. Sahl,

We are delighted to inform you that your manuscript, "Multiple phylogenetically-diverse, differentially-virulent </i>Burkholderia pseudomallei</i> isolated from a single soil sample collected in Thailand," has been formally accepted for publication in PLOS Neglected Tropical Diseases.

Best regards,

Shaden Kamhawi

co-Editor-in-Chief

Paul Brindley

co-Editor-in-Chief
